# DriveE2E: Benchmarking Closed-Loop End-to-End Autonomous Driving Based-on Real-World Traffic Scenarios

## Abstract

End-to-end learning has demonstrated considerable promise in advancing autonomous driving by fully leveraging sensor data. Recently, many end-to-end models have been developed, with a substantial number evaluated using the nuScenes dataset in an open-loop manner. However, open-loop evaluations, which lack interaction with the environment, fail to fully capture the driving capabilities of these models. While closed-loop evaluations, such as those using the CARLA simulator, allow for interaction with the environment, they often rely on rule-based, manually configured traffic scenarios. This approach leads to evaluations that diverge significantly from real-world driving conditions, thus limiting their ability to reflect actual driving performance. To address these limitations, we introduce a novel closed-loop evaluation framework that closely integrates real-world driving scenarios with the CARLA simulator, effectively bridging the gap between simulated environments and real-world driving conditions. Our approach involves the creation of digital twins for 15 real-world intersections and the incorporation of 800 real-world traffic scenarios selected from a comprehensive 100-hour video dataset captured with highly installed infrastructure sensors. These digital twins accurately replicate the physical and environmental characteristics of their real-world counterparts, while the traffic scenarios capture a diverse range of driving behaviors, locations, weather conditions, and times of day. Within this twinned environment, CARLA enables realistic simulations where autonomous agents can dynamically interact with their surroundings. Furthermore, we have established a comprehensive closed-loop benchmark that evaluates end-to-end autonomous driving models across these diverse scenarios. Notably, this is the first closed-loop end-to-end autonomous driving benchmark based on real-world traffic scenarios. Video demos are provided in the supplementary materials.

## 1 Introduction

End-to-end autonomous driving (E2EAD) has recently shown substantial advances and potential, exemplified by models like UniAD Hu et al. (2023) and Tesla's FSD V12 system Tesla Oracle (2024). Unlike traditional methods that optimize individual tasks in isolation and then integrate them through post-processing, the end-to-end approach directly optimizes the final planning output, thereby reducing error accumulation and information loss. E2EAD is also considered to fully exploit the potential of large datasets, making significant strides toward Level 4 autonomous driving.

Effective evaluation plays an essential role in the advancement of E2EAD research, especially in the era of the rapid emergence of new E2E algorithms. There are two primary evaluation ways for E2EAD systems. The first way, open-loop evaluation, mainly assesses the E2EAD's performance against pre-recorded expert driving route, like utilizing real-world nuScenes Caesar et al. (2020) datasets. In evaluation, the E2EAD system processes sensor data from a predefined route to predict future trajectories. However, this method cannot generate new observations based on the decisions of the ego vehicle. Consequently, open-loop evaluation often reduces to a trajectory prediction task Zhai et al. (2023); Li et al. (2024), which limits its assessment of vehicle-environment interaction and independent decision-making. The second way is closed-loop evaluation, which allows the ego vehicle to receive new observations based on its actions and offers a more realistic

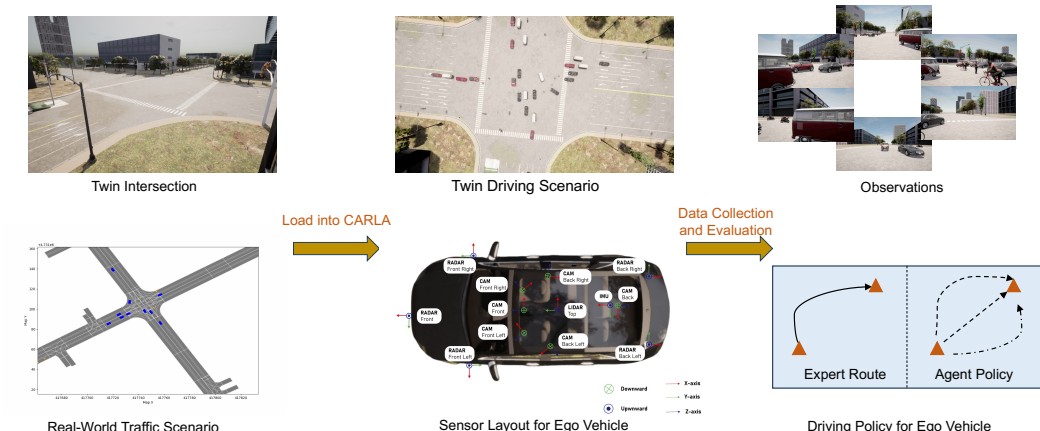

Figure 1: **Overview of DriveE2E.** We begin by constructing digital twins of real-world intersections and capturing corresponding traffic scenarios. These scenarios are then loaded into CARLA to create twin driving environments, with sensors equipped on the designated ego vehicle. Along the expert-defined route, we collect expert data for training E2EAD models. Using the agent policy output from the E2EAD systems, we evaluate their driving performance.

simulation and better reflection of the system's decision-making capabilities compared to open-loop evaluation. Since actual online vehicle testing is too expensive, the current closed-loop evaluation is mainly based on driving simulators. Existing closed-loop benchmarks, such as CARLA Leaderboard V2 CARLA Contributors (2024) and Bench2Drive Jia et al. (2024), also conduct the evaluation in the CARLA Dosovitskiy et al. (2017) simulator. However, in addition to rendering the scenario using simulation, the traffic scenario of the driving scenarios they used in the evaluation is also constructed using simulation or manual configuration. These traffic scenarios, often generated manually and randomly within constraints, can significantly deviate from real-world traffic situations. The significant discrepancies between the simulated and real-world traffic situations are mainly sourced from two aspects: 1) The behavior of traffic participants is heavily influenced by the actual road structure, but these manually generated traffic scenarios lack the relation with existing map topologies. 2) Interactions among traffic participants are crucial, yet the scenarios generated often lack realistic interactions. As a result, the evaluation results of these benchmarks may not accurately reflect real-world driving abilities.

To advance research in E2EAD and address the gap between real-world driving tests and simulation-based evaluations, we present DriveE2E, a closed-loop benchmark grounded in real-world traffic scenarios, with a particular emphasis on challenging urban intersections. The core innovation involves constructing digital twins of actual intersections and capturing real traffic scenarios from corresponding physical locations. These elements are integrated into the CARLA simulator to create high-fidelity digital twin driving scenarios. In this setup, a specifically designed ego vehicle, equipped with sensors, collects data rendered by CARLA and operates within the twin driving scenarios, guided by control commands generated by E2EAD algorithms. This twin design enables DriveE2E to provide comprehensive end-to-end evaluation capabilities for autonomous driving systems. Specifically, we constructed digital twins of 15 intersections located in urban Beijing, each featuring a variety of roads and topological structures to ensure a diverse range of traffic scenarios. From 100 hours of footage at these intersections, we selected 800 multi-view video clips and generated corresponding traffic scenarios that encompass eight driving behaviors, six weather conditions, and various times of day, ranging from morning to night. Each traffic scenario is richly detailed, including information such as trajectories of traffic participants, traffic light states, weather conditions, lighting, and vehicle IDs for the assignment of the ego vehicle. Notably, the multi-view videos are captured from high-positioned roadside cameras, which offer a broader field of view compared to typical vehicle-mounted sensors. This effectively alleviates the occlusion, allowing for comprehensive coverage of the intersection and ensuring the accurate capture of complete traffic flows. To ensure the DriveE2E benchmark can be utilized fairly and effectively by the research community, we collected 800 sensor data clips along the original driving routes, corresponding to 800 distinct driving scenarios. The dataset was divided into training, validation, and test sets in a 400:200:200

Table 1: Comparison with related planning evaluation benchmarks: DriveE2E is designed to minimize the evaluation gap between simulation and real-world on-road testing for closed-loop, end-to-end autonomous driving based on real-world traffic scenarios and twin driving scenarios.

| Benchmark | Year | Sensor | E2E | Closed-Loop | Driving Scenario | | |
| --- | --- | --- | --- | --- | --- | --- | --- |
| | | | | | Static Scene | Traffic Scenario | Rendering |
| Interaction Zhan et al. (2019) | 2019 | ✗ | ✗ | ✗ | Real | Real | - |
| Lyft Level 5 Houston et al. (2021) | 2021 | ✗ | ✗ | ✗ | Real | Real | - |
| nuScenes Caesar et al. (2020) | 2019 | ✓ | ✓ | ✗ | Real | Real | Real |
| Waymo Sun et al. (2020) | 2019 | ✓ | ✓ | ✗ | Real | Real | Real |
| Waymax Gulino et al. (2023) | 2023 | ✗ | ✗ | ✓ | Real | Real | - |
| nuPlan Caesar et al. (2021) | 2021 | ✓ | ✗ | ✓ | Real | Real | Real |
| CARLA LB V2CARLA Contributors (2024) | 2024 | ✓ | ✓ | ✓ | Twin | Sim | Carla |
| Bench2Drive Jia et al. (2024) | 2024 | ✓ | ✓ | ✓ | Twin | Sim | Carla |
| **On-Road Testing** | - | ✓ | ✓ | ✓ | Real | Real | Real |
| **DriveE2E (Ours)** | 2024 | ✓ | ✓ | ✓ | Twin | Real | Carla |

ratio. Specifically, 400 clips were designated for model training, while 200 clips were reserved for open-loop evaluation as a supplementary measure for E2EAD methods. Closed-loop evaluation was also conducted on the 200 validation scenarios to further assess performance.

**Notably, DriveE2E is the first twin-based, closed-loop benchmark for end-to-end autonomous driving grounded in real-world traffic scenarios. It is specifically designed to bridge the gap between simulation-based evaluations and real-world driving tests.** Our contributions can be summarized as follows:

- We developed a twin-based driving scenario solution for closed-loop evaluation in end-to-end autonomous driving, integrating real-world driving scenarios into the CARLA simulator. This approach reduces the gap between real-world driving tests and simulation evaluations, ensuring that the evaluation more accurately reflects real-world driving performance, making it highly valuable for current E2EAD research.

- We create 15 digital twin intersections and select 800 real-world traffic scenarios from a traffic database of 100-hour duration to develop twined driving scenarios. These digital twin intersections replicate the road and built elements of their real-world counterparts, while the traffic scenarios encompass diverse driving behaviors, locations, weather conditions, and time periods.

- We establish a comprehensive closed-loop benchmark for end-to-end autonomous driving on the diverse driving scenarios, evaluating four classic baseline E2EAD methods, including UniAD Hu et al. (2023), VAD Jiang et al. (2023), TCP Wu et al. (2022), and AD-MLP Zhai et al. (2023).

## 2 RELATED WORKS

**End-to-End Autonomous Driving.** End-to-end autonomous driving systems offer a compelling alternative to traditional modular designs by integrating perception, prediction, and planning into a single, differentiable model Hu et al. (2023); Chen et al. (2024a); Chib & Singh (2024). Unlike conventional modular methods that often struggle with the complexity of real-world scenarios, end-to-end approaches optimize the entire system holistically, directly processing raw sensor data into driving actions Jiang et al. (2023); Jia et al. (2023); Shao et al. (2024). Recent advancements have focused on utilizing transformers-based models Prakash et al. (2021); Chitta et al. (2023); Shao et al. (2023a); Jaeger et al. (2023); Shao et al. (2023b) and LLM-enhanced models Pan et al. (2024); Chen et al. (2024b); Xu et al. (2024); Fu et al. (2024); Sima et al. (2024), significantly enhancing the performance of these systems. These developments address key challenges such as generalization Wang et al. (2024) and interpretability Xu et al. (2024); Sima et al. (2024), leading to superior results on benchmarks for autonomous driving tasks.

**Evaluation Benchmarks for E2EAD.** In the context of E2EAD, benchmark evaluations play a crucial role as they provide standardized metrics for measuring progress and help assess the practical applicability and robustness of E2EAD systems. There are two primary methods for evaluating E2EAD algorithms. The first is open-loop evaluation, which utilizes metrics like L2 error and collision rate. This straightforward approach is widely used in E2EAD assessments Hu et al. (2022;

2023); Jiang et al. (2023); Chen et al. (2024c); Yu et al. (2024) but lacks interaction with the environment, limiting its ability to evaluate the algorithm's planning capabilities Zhai et al. (2023); Li et al. (2024). The second method is closed-loop evaluation, which typically relies on simulators to enable interaction between the ego vehicle and environmental agents. The most prominent end-to-end closed-loop simulators include CARLA Dosovitskiy et al. (2017), which has spawned several benchmarks like CARLA Leaderboard CARLA Contributors (2024), Longest6 Chitta et al. (2023), V2XVerse Liu et al. (2024), and Bench2Drive Jia et al. (2024). However, these benchmarks rely on artificially created scenarios rather than real-world trajectories. Other closed-loop evaluation platforms for autonomous driving planning, such as nuPlan Caesar et al. (2021) and Waymax Gulino et al. (2023), also exist but currently do not support end-to-end algorithm evaluation. In addition, there are some datasets, like Lyft Level 5 Houston et al. (2021) and Interaction Zhan et al. (2019), focus on the motion prediction task, which can be only used to test Non-E2E planning in an open-loop manner. Different from the existing benchmarks, Our DriveE2E is the first closed-loop E2EAD benchmark grounded in real-world traffic scenarios, which would enable a more realistic close-loop evaluation for E2EAD methods. We also provide these comparisons in Tab. 1.

## 3 DRIVEE2E

In this section, we introduce DriveE2E, the first benchmark designed specifically for evaluating end-to-end autonomous driving (E2EAD) systems using real-world traffic scenarios in a closed-loop approach. We start by highlighting the key features of DriveE2E. We then detail the process of creating the digital twins of real-world static traffic environments. Next, we provide descriptions of the expert data collection process for imitation-based model training and evaluation. Finally, we explain the methodology for evaluating E2EAD systems in a closed-loop manner using DriveE2E.

### 3.1 THE FEATURES OF DRIVEE2E

DrivieE2E contains 800 twined driving scenarios located in 15 intersection areas, covering a range of driving behaviors, weather conditions, and times of day from morning to night. Specifically,

- **Each twin intersection**, including road elements, traffic lights, and building elements on both road sides, is a digital twin of and consistent with the corresponding real-world intersection at Beijing city. We call this digital twin intersection as **Twin Intersection**. These twin intersections have diverse and complex road elements and topological structures, which help to evaluate the road understanding ability of the E2EAD systems. Visualization examples are provided in the appendix.

- **Each dynamic traffic scenario**, including traffic participants and their behaviors as well as traffic light signals, is sourced from real-world traffic data. These scenarios encompass a variety of elements such as pedestrians, non-motor vehicles, and cars, which are essential for evaluating the interactive capabilities of E2EAD systems within complex urban environments.

- **Each driving task** within a driving scenario is defined based on the original driving behaviors observed in real traffic scenarios, such as turning left while pedestrians are crossing. These tasks encompass a range of driving behaviors, which are crucial for assessing the driving capabilities of E2EAD systems. Detailed descriptions of these driving behaviors are also provided as follows.

**Driving Behaviors** DriveE2E identifies and categorizes 8 typical scenario types at intersections from 800 real-world traffic scenarios. These behaviors include Interaction with Pedestrians and Cyclists (IPC), Competing with Other Vehicles (COV), Passing through during Yellow Lights (YLW), Making a U-turn (UT), Stopping at Red Lights (STP), Going Straight through Intersection (STR), Making a Left Turn (LFT), and Making a Right Turn (RT). A detailed description of each driving behavior is provided below, and the distribution of these behaviors is illustrated in Fig.2(a).

- **IPC**: *Interaction with Pedestrians and Cyclists* involves safely navigating around or yielding to pedestrians and cyclists.

- **COV**: *Competing with Other Vehicles* refers to scenarios where the vehicle asserts its position in traffic, such as during merges or unprotected left turns.

- **YLW**: *Passing through during Yellow Lights* describes the decision-making process of whether to stop or proceed when the light turns yellow, balancing safety and timing.
- **UT**: *Making a U-turn* involves turning the vehicle to reverse its direction, either partially or fully, at an intersection or designated point.
- **STP**: *Stopping at Red Lights* involves halting the vehicle to comply with traffic signals.
- **STR, LFT, RT**: *Going Straight through Intersection, Making a Left Turn, and Making a Right Turn* are the most common driving behaviors at intersections, not specifically categorized under the other types.

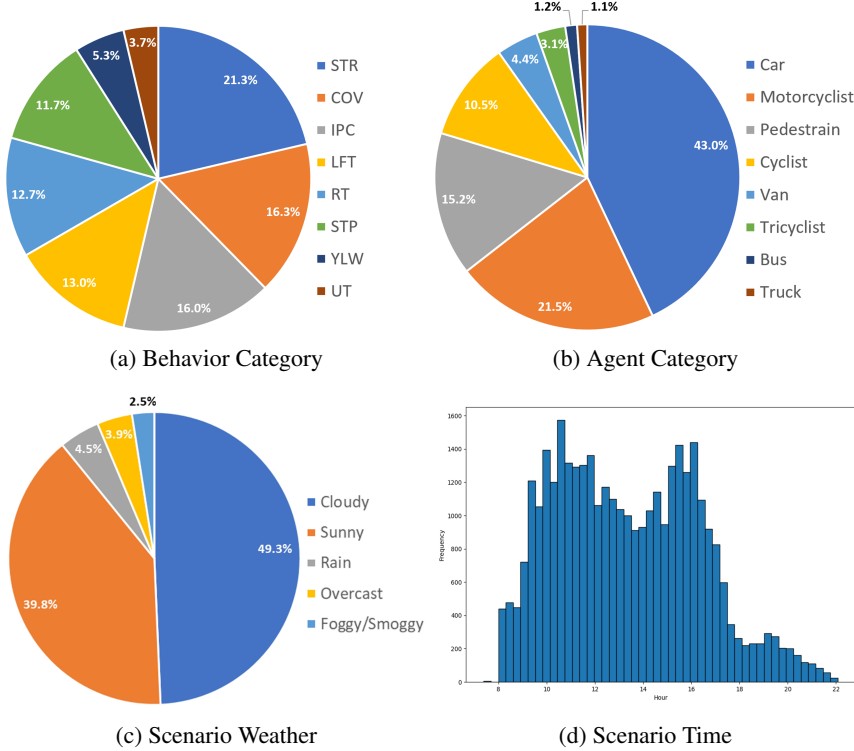

(a) Behavior Category         (b) Agent Category

(c) Scenario Weather        (d) Scenario Time

Figure 2: Data distribution of the driving scenarios

**Data Distributions**  The distributions of agent categories, weather conditions, and driving time are illustrated in Fig.2. As shown in Fig.2(b), DriveE2E includes 8 agent types, with the majority being cars, motorcycles, pedestrians, and cyclists, along with less common categories such as trucks, buses, tricyclists, and vans. Fig.2(c) demonstrates that DriveE2E encompasses 6 types of weather conditions, including uncommon ones like rain, overcast, and foggy weather. Fig.2(d) shows the time distribution of real trajectories in DriveE2E, which spans the entire day from morning to night, including peak hours when challenging scenarios are more likely to occur.

## 3.2 THE GENERATION OF TWIN DRIVING SCENARIOS

The generation of the twin driving scenarios in DriveE2E is mainly composed of three steps: **1) Twin Intersections Generation**: Creating digital twins of static intersections, which include complex road elements, including roadside infrastructures, such as traffic light poles, signs, lanes, crosswalks, stop lines, and surrounding buildings. **2) Dynamic Traffic Scenario Acquisition**: Collecting, annotating, normalizing and filtering dynamic real-world traffic scenarios to cover as many traffic conditions and driving behaviors as possible. These scenarios include traffic participants and their behaviors, and traffic light signals. **3) Loading and Configuring**: Loading dynamic traffic scenarios as well as their twin intersection into CARLA simulator, and configuring the appearance in the simulator. In the following parts, we will further explain how to create digital twins of static intersections and how to generate dynamic traffic scenarios.

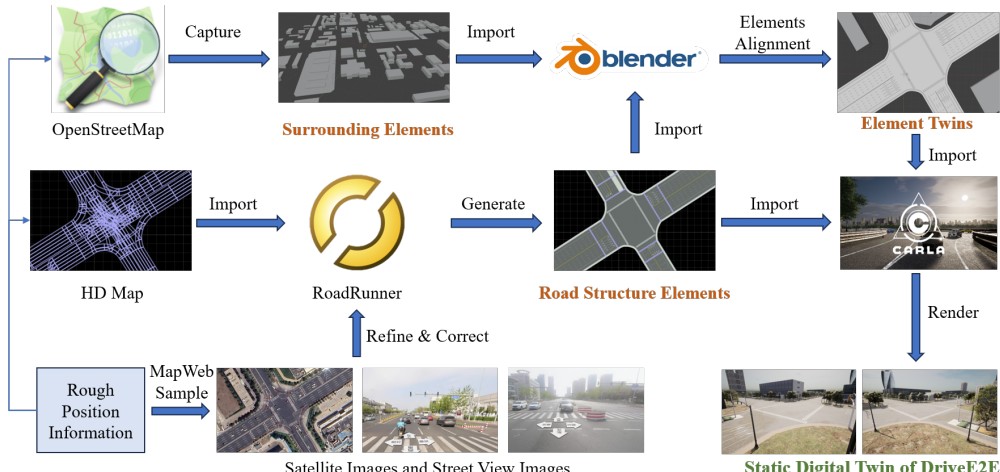

Figure 3: Twin Generation for Static Intersections: obtained HD Maps for intersections; refined structures in RoadRunner; collected data from OpenStreetMap; merged elements in Blender and rendered the whole static scenario in Carla.

**Twin Generation for Static Intersections.** We first obtained HD Maps for the areas covering the selected 15 intersections, organized similarly to Argoverse Chang et al. (2019). The location distribution of the selected 15 intersections is shown in Fig. 4(a). These HD Maps include lane centerlines, crosswalks, and stop lines, all represented as vector data. These maps were then loaded into RoadRunner[1], where we meticulously refined and corrected the road structure elements, ensuring accuracy by referencing high-resolution satellite images and street view images. Additionally, we employed OpenStreetMap[2] to gather information on surrounding elements, such as building data, and further configured the appearance attributes for these elements to ensure a more realistic and detailed representation of the intersection environments. The road structure and surrounding elements were then merged in Blender[3] to manually ensure accurate alignment of all elements. Finally, we completed the twin for each static intersection by incorporating all these elements into a unified simulation environment, capturing the intricate details necessary for realistic twins towards autonomous driving research.

**Dynamic Scenario Acquisition** Similar to the sensor deployment in DAIR-V2X Yu et al. (2022) and RCooper Hao et al. (2024), we first installed roadside cameras at each of the 15 intersections, positioned at elevated heights to cover the entire area, as shown in Fig. 4(b). We collected sensor sequence data over a 100-hour period at 10Hz, along with recording traffic light signals at the same frequency. Additionally, we obtained related weather and lighting from the weather system. Next, the collected sensor data were processed using trained 3D object detection Rukhovich et al. (2022) and tracking models Weng et al. (2020) to generate trajectory sequences encompassing over 1,000,000 annotated bounding boxes, each assigned a class label from 8 categories and a unique trajectory ID. We meticulously filtered and optimized these trajectories to form a high-quality traffic scenario database. From this database, we manually selected the ego vehicles and further classified their driving behaviors, which ensured that the scenarios accurately represented various driving behaviors. We then used these scenarios to build DriveE2E, selecting 800 scenarios to ensure a diverse range of scenes and driving conditions.

## 3.3 EXPERT DATA COLLECTION

To ensure that the DriveE2E benchmark is utilized fairly and effectively by the research community, we release observation data for the 800 constructed scenarios to facilitate the training of imitation-

---

[1]RoadRunner MathWorks (2023): a 3D environment editing tool used for designing and editing road and traffic scenes for simulation and testing of autonomous driving systems.

[2]OpenStreetMap OSM contributors (2023): a global, user-collaborative, open, and free map database.

[3]Blender Blender Studio (2023): a free 3D creation software for modeling, animation and rendering.

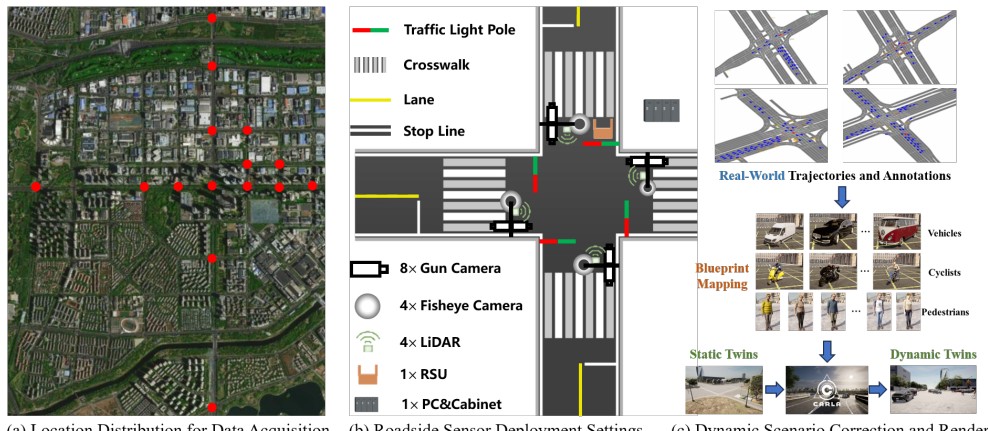

(a) Location Distribution for Data Acquisition    (b) Roadside Sensor Deployment Settings    (c) Dynamic Scenario Correction and Render

Figure 4: Twin Generation for Dynamic Intersections. (a) Location Distribution for Data Acquisition: The locations for dynamic data collection correspond to the 15 specified static intersections in Beijing, China. (b) Roadside Sensor Deployment Settings: Sensor sequence data is collected alongside traffic light signal recordings. (c) Dynamic Scenario Correction and Rendering: Sensor data is auto-annotated to generate trajectories, followed by manual corrections, mapping real-world actors to blueprints, and importing them into Carla for rendering and simulation.

Table 2: Key Sensor Specifications for expert data acquisition.

| Sensor | Details |
|---|---|
| 1x LiDAR | 64 channels, 85-meter range, $360°$ horizontal FOV, $+10°$ to $-30°$ Vertical FOV |
| 6x Camera | Surround coverage, RGB, 900x1600 resolution, JPEG compressed |
| 5x Radar | 100-meter range |
| 1x IMU&GPS | Position, heading, speed, acceleration, and angular velocity |

learning-based E2EAD methods. We have collected and saved the sensor data and 3D annotations from the view of ego vehicle as **Expert Data**. Specifically, we drove an ego vehicle along its original **real-world route** as mentioned in dynamic scenario acquisition, equipped with sensors similar to those used in nuScenes Caesar et al. (2020) and Bench2Drive Jia et al. (2024), as shown in Fig.1. Sensor specifications are detailed in Tab.2. The sensor data were recorded at 10Hz, totaling 800 clips, corresponding to the 800 scenarios mentioned. For data partitioning, 400 clips are designated as training data, 200 clips are designed for evaluation, and 200 clips are reserved for testing.

### 3.4 CLOSED-LOOP EVALUATION

Closed-loop evaluation for E2EAD allows an autonomous vehicle to interact with and respond to dynamic changes in real-time. This method continuously updates the traffic environment observations based on the autonomous system's decisions, enabling a comprehensive assessment of its decision-making capabilities.

In DriveE2E, the autonomous vehicle is tasked with successfully navigating from the source location $(x_{src}, y_{src})$ to the destination location $(x_{dst}, y_{dst})$ within a driving scenario. The source and destination locations correspond to the vehicle's positions in its original driving route. The E2EAD system receives raw sensor data (including multi-view images and point clouds), GPS coordinates, and target waypoints as inputs. These waypoints are obtained by downsampling the vehicle's original route. The output of the system should be control commands, such as steering angle, throttle, and brake. Alternatively, the output could be future planning waypoints, which are then converted into control commands using the CARLA simulator.

**Evaluation Metrics.** Here we adopt three metrics to evaluate the performance of the E2EAD system, following CARLA LB V2 CARLA Contributors (2024) and Bench2Drive Jia et al. (2024):

Table 3: Open-Loop and Closed-Loop Evaluation Results of E2EAD Methods in DriveE2E. Considering that UniAD has not yet converged, we have not reported its closed-loop results yet.

| Methods | Open-Loop Metric ↓ | | | Closed-Loop | |
|---|---|---|---|---|---|
| | 1s | 2s | Average | SR (%) ↑ | DS ↑ |
| AD-MLP Zhai et al. (2023) | 4.82 | 10.48 | 7.65 | 6.85 | 8.94 |
| UniAD Hu et al. (2023) | 0.70 | 1.58 | 1.14 | - | - |
| VAD Jiang et al. (2023) | 0.58 | 1.10 | 0.84 | 45.14 | 55.15 |
| TCP Wu et al. (2022) | 1.60 | 3.53 | 2.57 | 7.42 | 10.47 |

- **Success Rate (SR).** This metric measures the percentage of successfully completed routes within a certain time. There should not be any conflicts or traffic violation, such as not leaving the road area, during the driving process.

- **Driving Score (DS).** This metric measures the driving performance while taking the route completion $RC_i$ and infraction penalty of $i$-route into account as Eq. 1.

$$DS = \frac{1}{n_{total}} \sum_{i=1}^{n_{total}} RC_i * \prod_{j=1}^{inf_i} (p_i^j), \tag{1}$$

where $n_{total}$ denotes the total number of routes, $inf_i$ means a set of infraction that the ego vehicle triggered in $i$-route, and $p_i^j$ denotes the infraction penalty coefficient. For more details about infraction types and coefficients, refer to CARLA LB V2.

## 4 EXPERIMENTS

### 4.1 BASELINES AND DATASETS

We implemented several classical End-to-End Autonomous Driving (E2EAD) models as baselines on the DriveE2E platform, using imitation learning for training. Specifically, we divide the 800 expert data clips collected into training, validation, and test sets in a 4:2:2 ratio, ensuring a balanced distribution of behavior categories and weather conditions in each set. The 400 training clips were used to train the models on A100 GPUs. We evaluated the trained models in a closed-loop setup in the validation set. In addition, open-loop evaluations were conducted on the same validation set to further assess performance. We report the performance of the model in terms of L2 error (m).

- UniAD Hu et al. (2023) employs queries to integrate key tasks such as perception, mapping, prediction, and planning. The standard training process for UniAD typically involves three stages. To accelerate training and reduce GPU resource consumption, we bypassed the initial stages by directly training the stage-2 model using the bevformer Li et al. (2022) model provided by Bench2Drive Jia et al. (2024) as a pre-trained model. We train UniAD for one epoch. It is important to note that these settings may lead to a reduction in UniAD's accuracy.

- VAD Jiang et al. (2023) employs Transformer queries while enhancing efficiency through a vectorized scene representation. We trained the VAD model for two epochs, using a pre-trained model provided by Bench2Drive Jia et al. (2024) as the pretrained model.

- AD-MLP Zhai et al. (2023) adopts a simple strategy by entering the past states of the ego vehicle into an MLP to generate future trajectory predictions. We train AD-MLP for 60 epochs.

- TCP Wu et al. (2022) predicts both trajectories and control signals. It only uses front-facing cameras and the ego state as inputs. Note that we did not train an expert model and did not use expert feature distillation during TCP training. TCP was trained for 27 epochs.

### 4.2 MAIN RESULTS

We present the evaluation results in Tab. 3, which include both the open-loop evaluation results (L2 error) and the closed-loop evaluation results (success rate and driving score).

**Open-loop Evaluation Results.** As shown in Tab. 3, AD-MLP exhibits a significantly high L2 error, with an average error reaching 7.65 m. This result contrasts with the performance observed on nuScenesCaesar et al. (2020); Zhai et al. (2023), where using only past ego status produced strong planning outcomes. The discrepancy is understandable, as DriveE2E incorporates a wider range of driving behaviors (Fig. 2), unlike nuScenes, where most behaviors are relatively straightforward. This highlights the increased challenge DriveE2E presents for driving evaluation. Both UniAD and VAD outperform AD-MLP and TCP, which is expected given that our benchmark is more challenging, and UniAD and VAD are specifically designed for planning tasks. While VAD achieves a lower L2 error than UniAD, it is premature to conclude that VAD performs better. UniAD was only trained for one epoch due to time constraints, and it has not yet fully converged.

**Closed-loop Evaluation Results.** Both AD-MLP and TCP exhibit very low success rates and driving scores, with AD-MLP achieving a 6.85 SR and 8.94 DS, and TCP achieving a 7.42 SR and 10.47 DS. In contrast, VAD performs considerably better in the closed-loop evaluations, with a 45.14 SR and 55.15 DS. These results indicate that relying solely on past ego status is insufficient for generating effective planning outputs in complex traffic environments.

**Relationship between Close-loop and Open-loop Evaluation Results.** To some extent, open-loop and closed-loop evaluations are related. For example, AD-MLP, which has the highest L2 error, also exhibits the worst driving performance in closed-loop evaluation. Conversely, VAD performs well in both open-loop and closed-loop assessments. This suggests that open-loop evaluations with difficult and diverse driving scenarios can provide insight into driving ability. However, the results across different methods do not always show a strictly consistent pattern between open-loop and closed-loop evaluations. This is because open-loop outputs do not necessarily correlate positively with the outcomes of closed-loop evaluations, which involve interaction. Therefore, closed-loop evaluation remains essential for accurately assessing driving ability.

### 4.3 PERFORMANCE ON DIFFERENT BEHAVIORAL SCENARIOS

We also evaluated all trained E2EAD systems across the eight different behavior categories in DriveE2E, with the results presented in Tab. 4. The performance of E2EAD systems in certain categories, such as IPC and COV, is worse compared to the STR category. This is because scenarios like IPC and COV involve interactions with other traffic participants, such as pedestrians and motor vehicles, which place greater demands on driving ability. In contrast, behaviors like going straight (STR) are simpler and require relatively lower driving skill.

Table 4: Close-loop Evaluation for Different Behavioral Scenarios.

| Models | Driving Score for Different Behavior Categories ↑ | | | | | | | |
|---|---|---|---|---|---|---|---|---|
| | COV | IPC | UT | YLW | STR | LFT | RT | STP |
| AD-MLP | 3.12 | 7.14 | 20 | 6.66 | 12.12 | 4.34 | 0 | 11.11 |
| VAD | 37.50 | 32.14 | 40.00 | 46.66 | 48.48 | 47.82 | 42.85 | 72.22 |
| TCP | 6.25 | 7.14 | 20.00 | 6.66 | 9.09 | 0.00 | 0.00 | 22.22 |

### 4.4 COMPARISON WITH OTHER CARLA-BASED SIMULATORS

We also compared the closed-loop evaluation results on our DriveE2E platform with those from Bench2Drive. The performance of different methods is generally consistent across both platforms. Notably, VAD performs better on DriveE2E, suggesting that the scenarios in DriveE2E are generally simpler than those in Bench2Drive. This is expected, as Bench2Drive intentionally includes many corner cases. In future work, we plan to incorporate more rare and challenging scenarios into DriveE2E.

## 5 CONCLUSIONS

This work presents DriveE2E, an innovative closed-loop benchmark aimed at advancing End-to-End Autonomous Driving (E2EAD) research by bridging the gap between simulation and real-world on-

Table 5: Comparison of Close-loop Evaluation Results with Other Benchmarks

| Models | Driving Score for Different Benchmarks ↑ | |
| --- | --- | --- |
| | Bench2Drive | DriveE2E |
| AD-MLP | 9.14 | 8.94 |
| UniAD | 37.72 | - |
| VAD | 39.42 | 55.15 |
| TCP | 23.63 | 10.47 |

road testing. By integrating real-world traffic scenarios into digital twin environments within the CARLA simulator, DriveE2E offers a realistic evaluation framework that overcomes the limitations of both traditional open-loop methods and existing CARLA-based closed-loop evaluations. The benchmark includes digital twins of 15 diverse urban intersections and 800 traffic scenarios encompassing various driving behaviors, weather conditions, and times of day. Additionally, we present a robust evaluation benchmark featuring four classic E2EAD methods, enabling comprehensive closed-loop assessments. This benchmark not only enhances the accuracy of performance evaluations but also improves the real-world applicability of E2EAD systems.

**Limitations and Future Work.** Currently, interactions with other traffic participants in both DriveE2E and the mainstream CARLA framework are very weak. We plan to enhance this by integrating a more advanced interaction controller in the future. There is still a big gap between the real data and the simulated data with the rendering based on the CARLA simulation. We are considering the incorporation of generative models to further increase the realism of the visual output.

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

APPENDIX

## I   VISUALIZATION OF THE TWINED INTERSECTIONS

DriveE2E develops digital twins for 15 static intersections, which include intricate roadside infrastructures, such as traffic light poles, signage, lanes, crosswalks, stop lines, and nearby buildings. The constructed twined intersections are illustrated in Fig. 5.

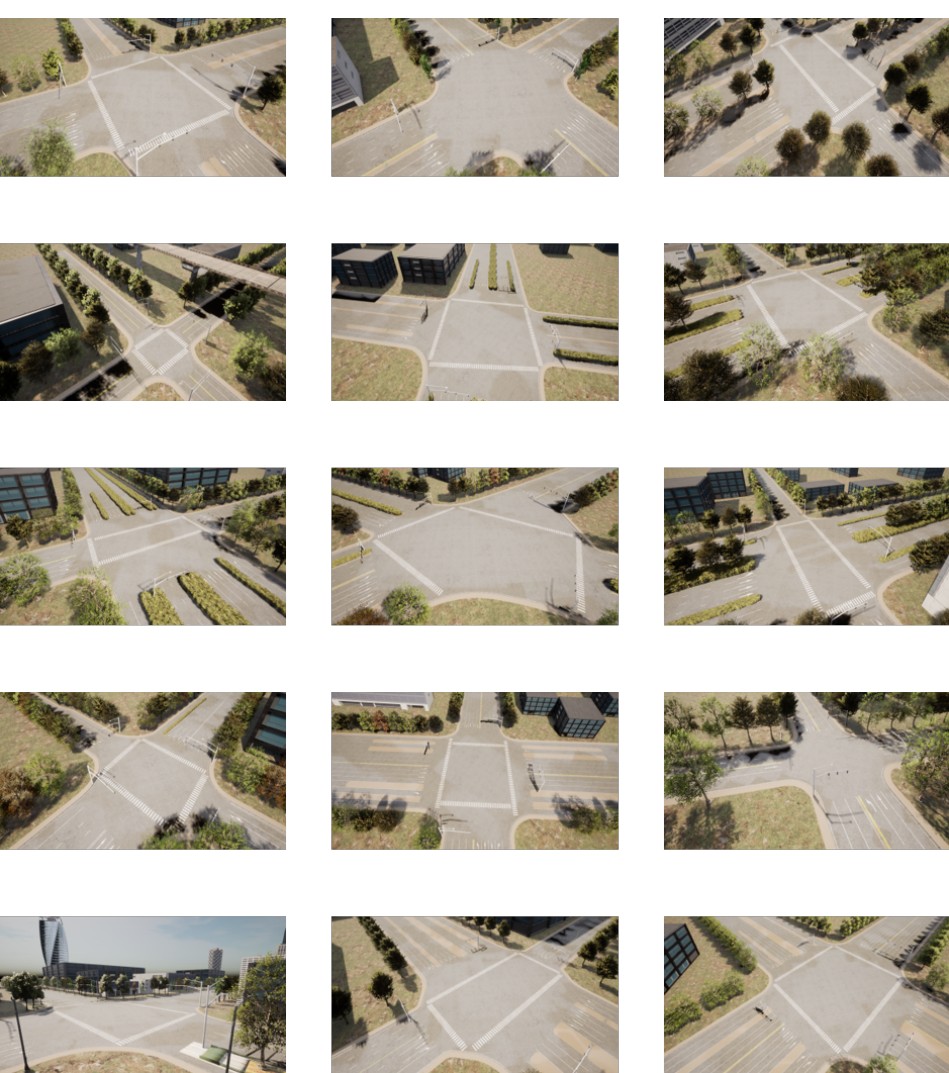

Figure 5: Digital twins of 15 static intersections, showcasing complex roadside infrastructures, including traffic light poles, signage, lanes, crosswalks, stop lines, and nearby buildings.

## II   ILLUSTRATION OF THE DRIVING SCENARIOS

**Driving Behavior Illustration.**   DriveE2E identifies and categorizes eight distinct driving scenarios from 800 clips of real-world traffic situations, capturing typical driving behaviors at intersections. The specific scenarios include Interaction with Pedestrians and Cyclists (IPC), Competing with Other Vehicles (COV), Passing through during Yellow Lights (YLW), Making a U-turn (UT), Stopping at Red Lights (STP), Going Straight through Intersection (STR), Making a Left Turn (LFT), and Making a Right Turn (RT). These eight scenarios are **further refined into 14 specific sub-scenarios** according to the condition of turning and anomaly. We illustrate these sub-scenarios in Fig. 6, Fig. 7 and Fig. 8.

**Twinning of Weather and Light Conditions.**   Thanks to effective dynamic scenario acquisition, DriveE2E accurately replicates the original weather and lighting conditions of the real-world scenarios. We collected weather data and timestamps during capture, allowing us to recreate the actual weather states and lighting angles in CARLA's weather system. To visually illustrate the effects of weather and lighting, we present one reconstructed scene under various weather and lighting conditions in Fig. 9.

### III   ILLUSTRATION OF BENCHMARK METHODS IN DRIVEE2E

This section primarily visualizes the performance of benchmark methods on DriveE2E. Due to space limitations, we present the successful and failed cases of the VAD model in three scenarios (COV, LFT, STR) in Fig. 10, Fig. 11, and Fig. 12.

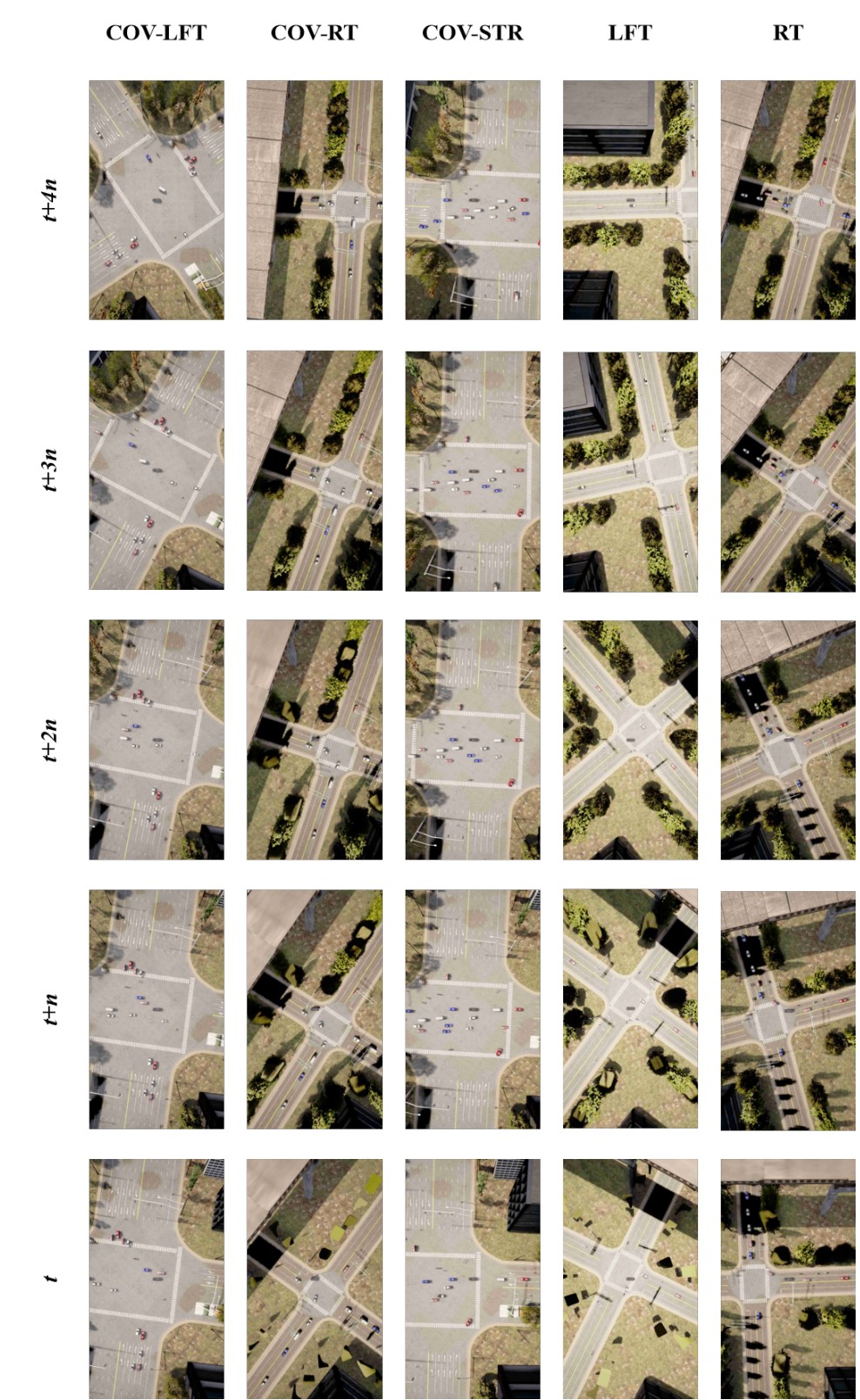

Figure 6: Driving Behavior Illustration (a) features five sub-scenarios: Competing with other vehicles while turning left (COV-LET), turning right (COV-RT), and going straight (COV-STR), along with normal left turns (LFT) and right turns (RT). Clear visualizations include serial RGBs in the **top-down view**, with the ego vehicle (in gray) positioned at the center of each image.

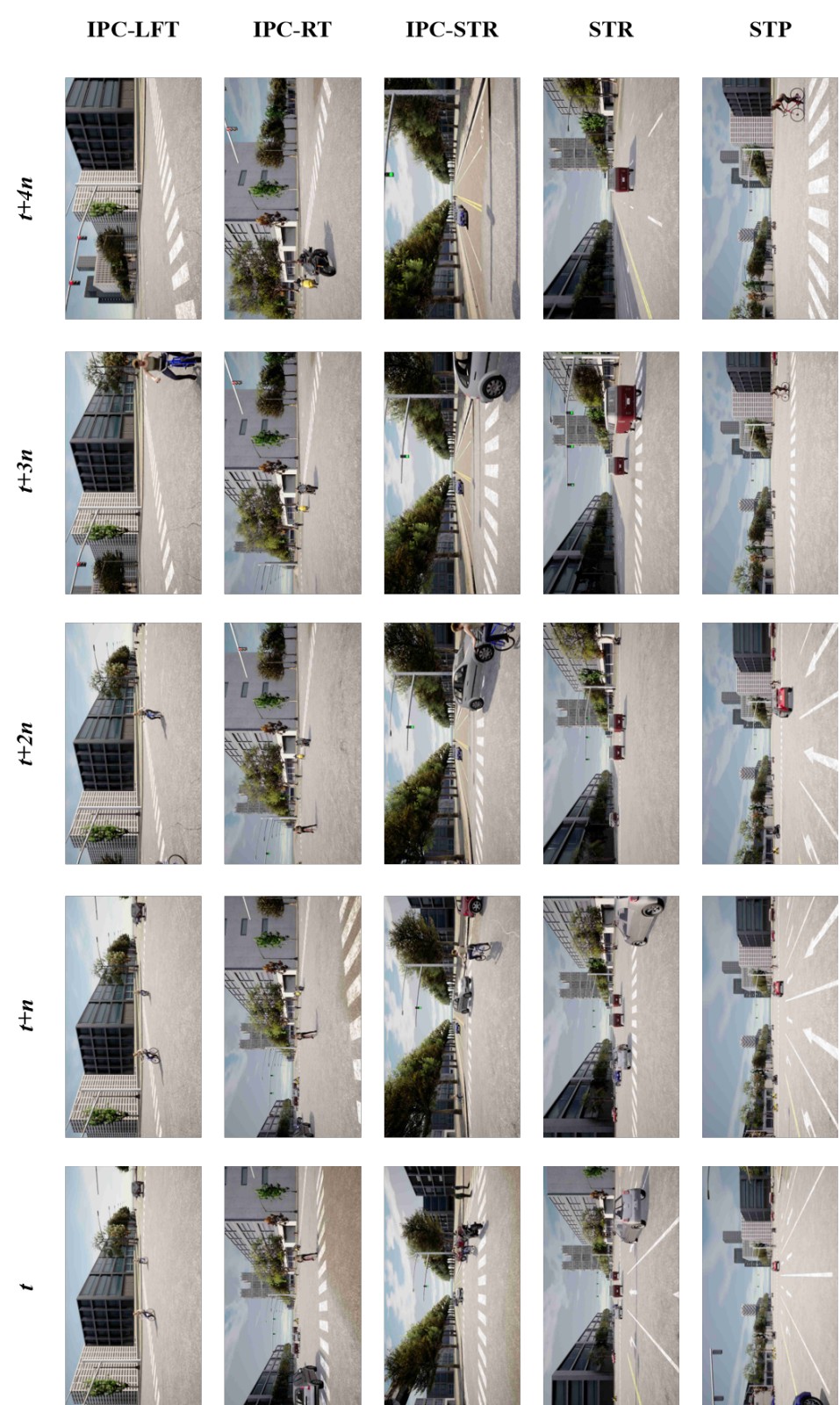

Figure 7: Driving Behavior Illustration (b) features five sub-scenarios: Interaction with pedestrians and cyclists while turning left (IPC-LET), turning right (IPC-RT), and going straight (IPC-STR), along with normal straight driving (STR) and stopping at red lights (STP). Clear visualizations include serial RGBs in the **forehead view**.

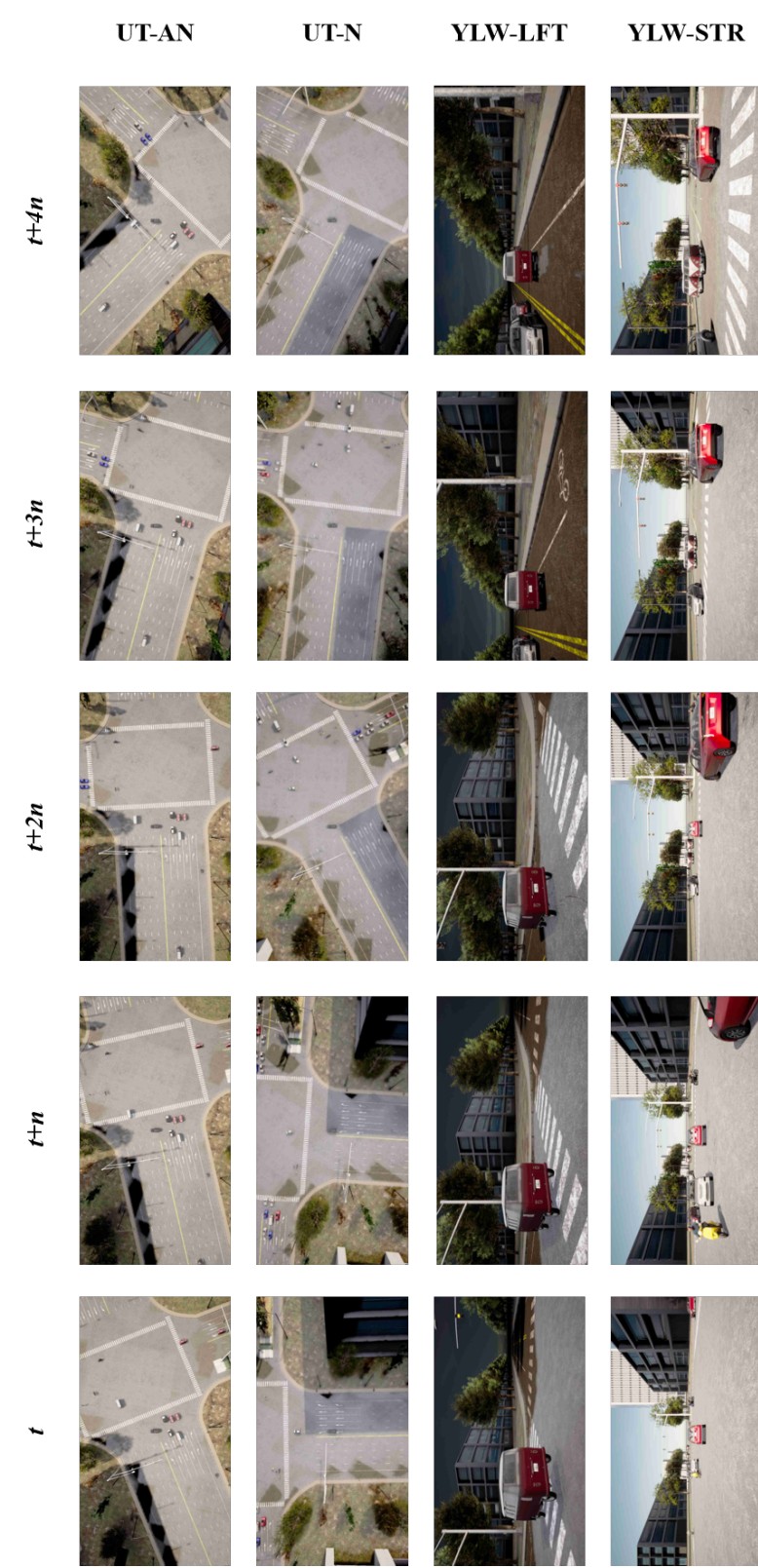

Figure 8: Driving Behavior Illustration (c) features four sub-scenarios: U-turns in abnormal (UT-AN) and normal conditions (UT-N), and passing through yellow lights while turning left (YLW-LFT) or going straight (YLW-STR). Clear visualizations include serial RGBs in both **top-down and forehead views**, with the ego vehicle (in gray) positioned at the center of each top-down image.

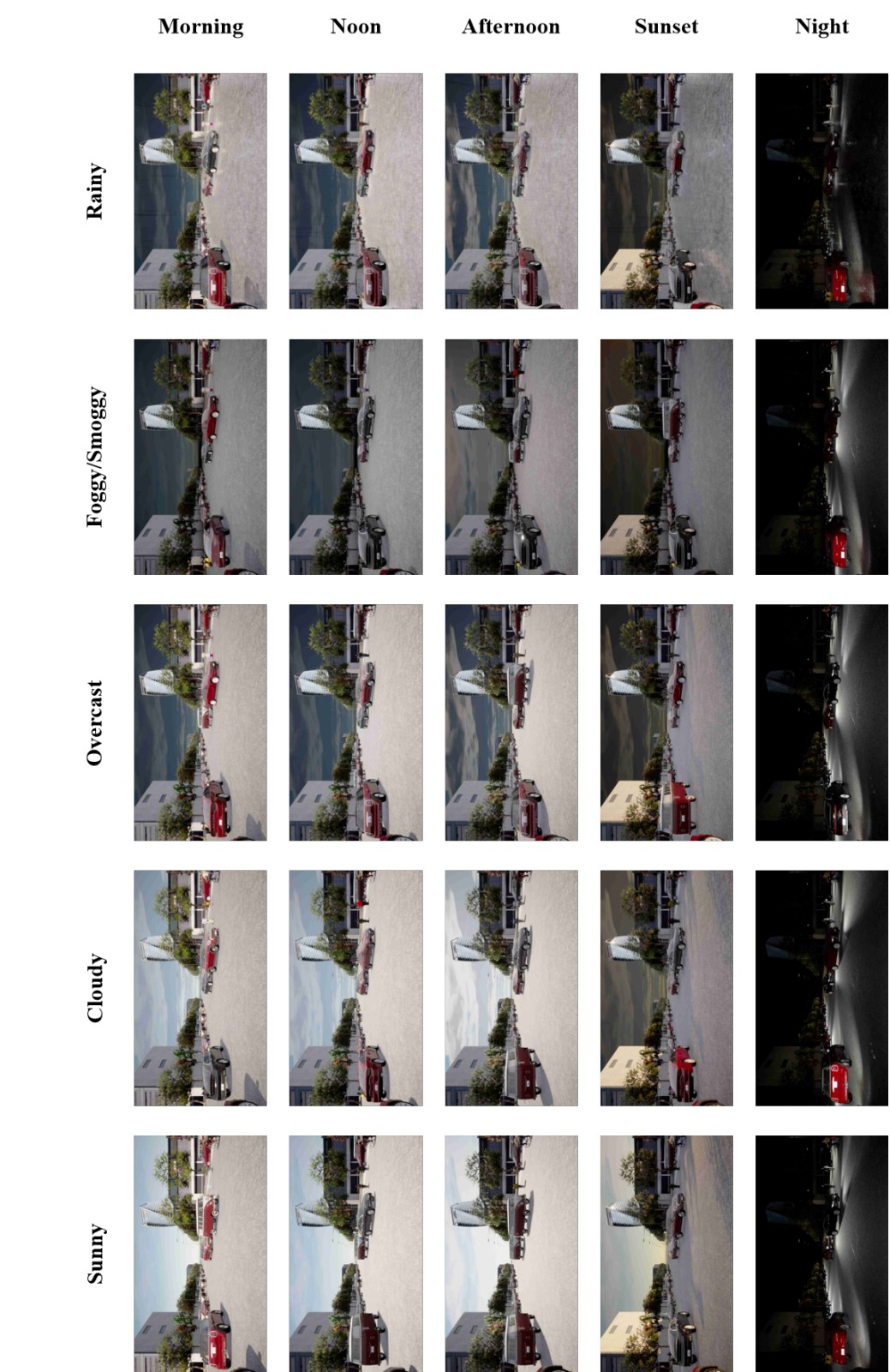

Figure 9: Twinning of Weather and Light Conditions. We present a reconstructed scenario under different weather and lighting conditions. The complex perceptual environment, including shadows and reflections on rainy days, has been effectively recreated.

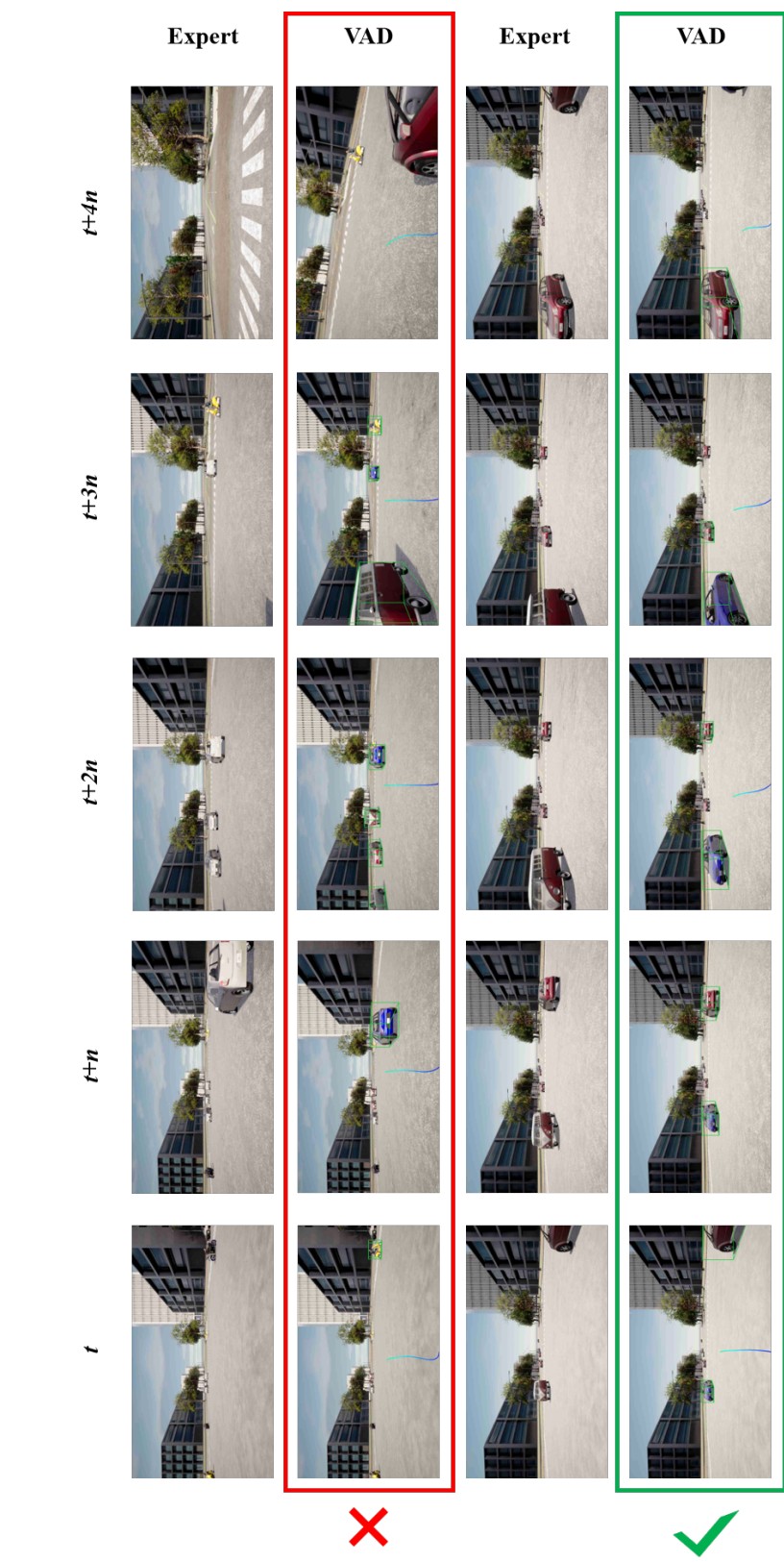

Figure 10: Successful and failed cases of the VAD model in the COV scenario. In the failed case, the VAD ego vehicle was overly cautious while competing for the lane with another vehicle, neglecting a car approaching from the right rear, which resulted in a collision due to its slow speed. In contrast, the successful case demonstrated a reasonable speed, with no collisions occurring.

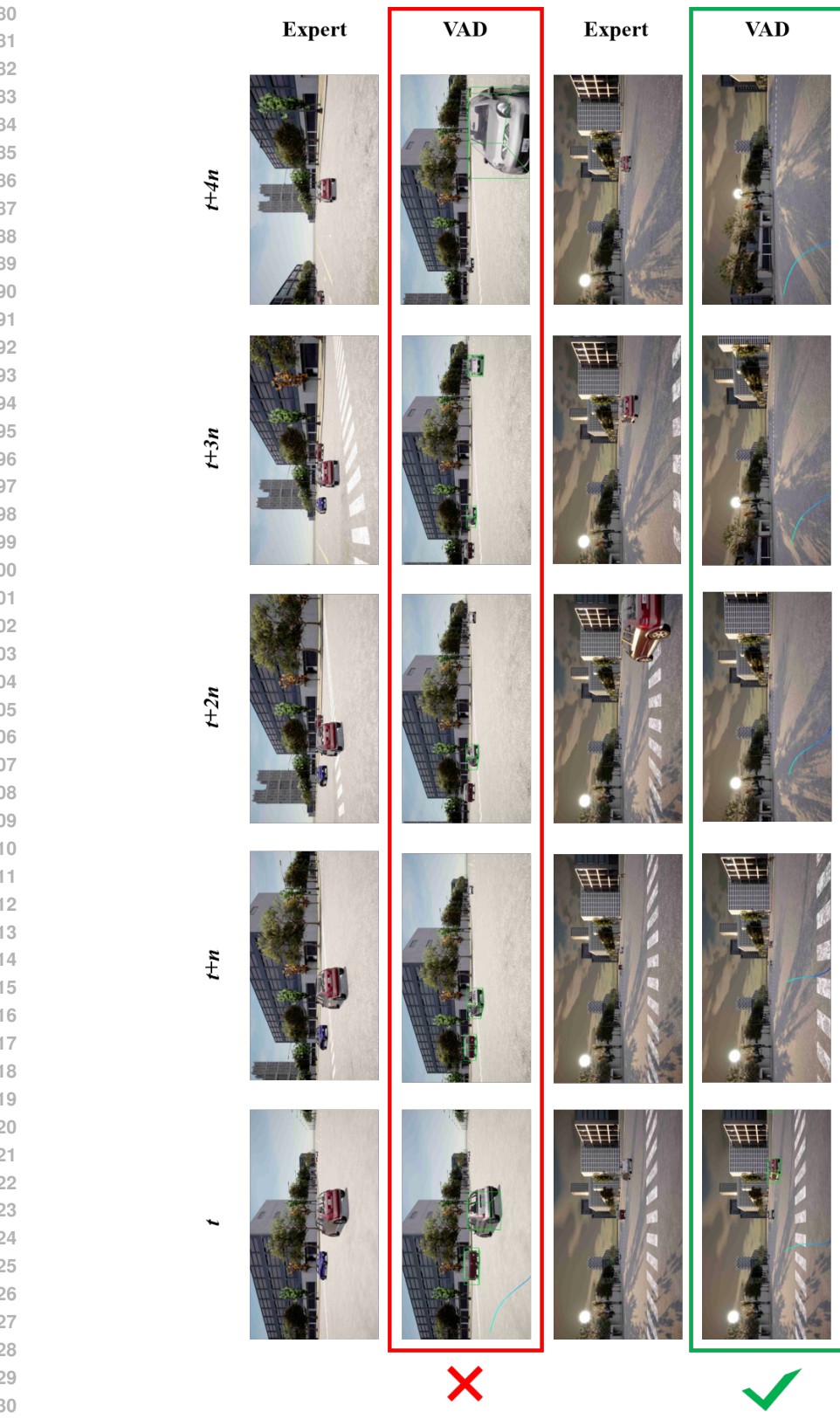

Figure 11: Successful and failed cases of the VAD model in the LFT scenario. In the failed case, the VAD ego vehicle was overly cautious during a left turn and failed to effectively predict oncoming traffic, leading to a collision. In contrast, the successful case saw the VAD ego complete its intended maneuver without interference from oncoming vehicles.

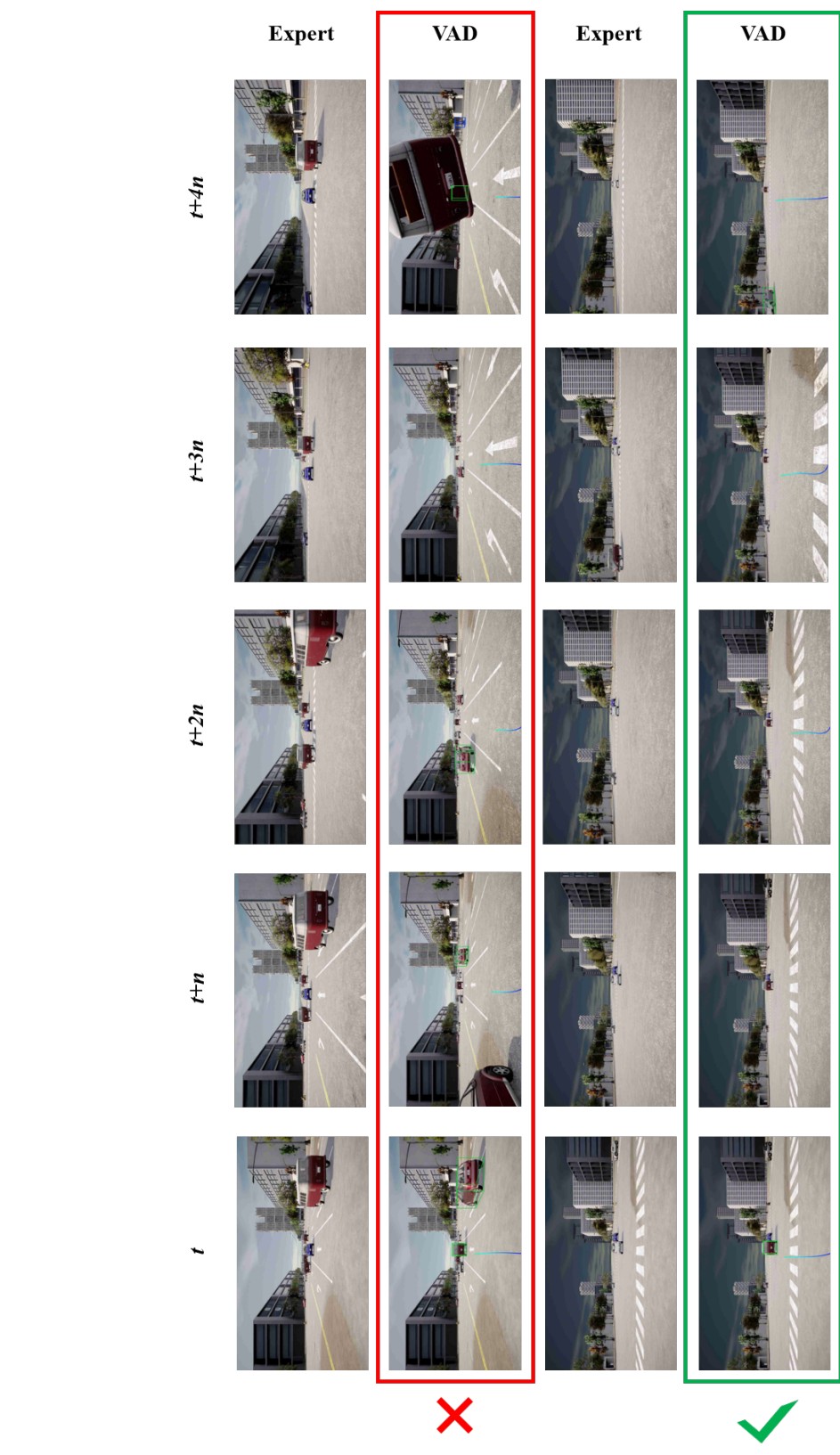

Figure 12: Successful and failed cases of the VAD model in the STR scenario. In the failed case, the VAD ego vehicle accelerated too slowly while moving straight, resulting in a collision with a trailing vehicle. In contrast, the successful case showed the VAD ego navigating the intersection at a reasonable speed.

