# OpenReview forum: "DriveE2E: Benchmarking Closed-Loop End-to-End Autonomous Driving Based-on Real-World Traffic Scenarios"
_ICLR.cc/2025/Conference — ICLR 2025 Conference Withdrawn Submission_

### Official Review · Reviewer_88qr · 2024-10-24

**Soundness:** 1
**Presentation:** 2
**Contribution:** 2
**Rating:** 3
**Confidence:** 5

**Summary:**

This work transforms real world data into digital twins in CARLA and run closed-loop evaluation with open sourced end-to-end baselines.

**Strengths:**

1. The extra digital towns and traffic scenes might have some use

**Weaknesses:**

1. This work claims to curate a digital twins in CARLA, which is a dataset & benchmark work.  Thus, I believe it is fairly well to share the anonymous link regarding the dataset and benchmark code. Otherwise, it is difficult to evaluate the contributions. Additionally, the authors does not mention the plans of open source, which limites its impact as a dataset & benchmark work.

2.  From the  visualization in supplymental materials, the quality of digital twins is much lower compared to latest CARLA (For example, Town12 and Town13) in terms of material diversity and map size.  Additioanlly, if there is no open source materials, the technical contributions are limited, as it is not new to create digital twins in CARLA. For example, CARLA town15 is a digital twins of University of Barcelona: https://carla.readthedocs.io/en/0.9.15/map_town15/. As a result, its value to the community might be limited.

**Questions:**

1. Do the authors plan to open source? Do the authors plan to open source the tutorial and curation code of digital twins?

2. Are other actors reactive to the ego vehicle? Or they could only follow the action in their log?

3. Does UniAD have a closed-loop results now?


In summary, I do not think barely describe the curation of digital twins from some real world data in CARLA, especially a rather navie ones compared to latest CARLA has enough contributions. Not to mention that there are no clues at all regarding the open source. I will maintain my reject rating if there is no data and code provided.

---

### Official Review · Reviewer_Wphb · 2024-11-01

**Soundness:** 3
**Presentation:** 2
**Contribution:** 1
**Rating:** 3
**Confidence:** 5

**Summary:**

This paper introduces DriveE2E, a benchmark for closed-loop evaluation of end-to-end autonomous driving (E2EAD) systems. By integrating real-world traffic scenarios into digital twin environments within the CARLA simulator, DriveE2E bridges the gap between simulated and real-world testing, enhancing evaluation accuracy. The benchmark includes digital twins of 15 intersections and 800 traffic scenarios, covering diverse conditions, to support comprehensive assessments of E2EAD performance​.

**Strengths:**

1. The authors have developed a comprehensive closed-loop simulation pipeline utilizing the CARLA platform. By incorporating real-world traffic data, they construct digital environments that facilitate closed-loop testing of end-to-end driving models. The engineering efforts appear solid, and I appreciate the authors for their thorough work.
2. The authors perform closed-loop testing with several state-of-the-art end-to-end models within the simulator, effectively demonstrating the simulator's capabilities and validating its effectiveness.
3. The exploration of the correlation between open-loop metrics and closed-loop performance is particularly insightful and adds valuable depth to the discussion.

**Weaknesses:**

1. While the engineering work is commendable, the paper lacks novelty. The concept of building digital twins using real-world traffic data is well-established. For example, MetaDrive [1] has implemented a similar approach, utilizing a larger dataset and integrating learning-based traffic agent control, which appears more advanced.
2. The process of importing traffic agents into CARLA is unclear. Additionally, during closed-loop testing, it is uncertain how these agents interact with the self-driving vehicle. If the interaction relies on CARLA's traffic manager, this raises concerns about the realism of the digital twin, as CARLA’s traffic manager may not accurately simulate real-world scenarios.
3. The paper does not validate the value of the digital town. It seems feasible to use CARLA's default environments and traffic flows for closed-loop testing, potentially yielding similar results. The authors should consider establishing metrics to demonstrate the enhanced realism and effectiveness of their digital town.
4. The simulator presented in the paper does not appear to be specifically tailored for end-to-end driving applications. It seems more like a generic simulator, which detracts from the overall narrative and impact of the paper.
[1] Metadrive: Composing diverse driving scenarios for generalizable reinforcement learning

**Questions:**

1) Novelty and Comparison: Can the authors clarify how their approach differs from existing work, such as MetaDrive? What specific advancements or unique contributions does this paper offer in comparison to similar systems?
2) Traffic Agent Interaction: How are the traffic agents integrated into CARLA, and how do they respond to the self-driving vehicle during closed-loop testing? Are these interactions governed by CARLA's traffic manager, and if so, what measures have been taken to ensure realistic behavior?
3) Validation of Digital Towns: What methodologies or metrics do the authors propose to validate the effectiveness and realism of their digital town compared to CARLA’s default environments? How can they demonstrate that their implementation yields more accurate or useful results?
4) End-to-End Driving Focus: In what ways does the simulator specifically cater to the needs of end-to-end driving models? Could the authors elaborate on how their simulator's design is tailored for this application, rather than being a more generic platform?

---

### Official Review · Reviewer_aC4H · 2024-11-03

**Soundness:** 3
**Presentation:** 3
**Contribution:** 2
**Rating:** 5
**Confidence:** 4

**Summary:**

In this work, 15 real-world traffic scenarios are meticulously reconstructed for closed-loop simulation in CARLA. Further, the original data clips corresponding to driving scenarios captured within these scenarios are released (or at least, promised to be released, there are no samples included in the supplementary material). A set of E2E AD models are additionally evaluated in open- and closed-loop on the benchmark, setting baseline values for future research.

**Strengths:**

Closed-loop simulation and evaluation is a timely topic in the AV research community, and this work correctly identifies many of the difficulties with current simulators and benchmarks.

This work brings real-world scenarios into simulation for E2E AV stacks to evaluate on, which is a much more realistic evaluation scheme than prior approaches.

The paper is written clearly and the core ideas are easy to follow.

It is clear a lot of work went into the development of this paper, and the broader community will certainly be able to leverage it.

**Weaknesses:**

A core weakness is the question of how "closed-loop" this closed-loop simulation is. It seems that agents are only replay and cannot interact with the ego-vehicle, which makes the closed-loop contribution of this paper primarily the CARLA scenarios and AV data.

There is also a question of if this work would be better received at a robotics or autonomous vehicle conference (or even as a technical report accompanying code release) compared to presentation at ICLR.

There are no data samples released for review. This is critical for a paper whose core contribution is a dataset release, and is the primary reason for the rejection-leaning rating. One samples are provided, I would be happy to raise my score (provided they are of the high-quality claimed in the paper).

**Questions:**

How accurately-captured is the static environment in simulation? For example, billboards, buildings, lights at night-time, etc. These factors play an important role in perception realism, but it is impossible to tell without the presence of the accompanying driving sensor data.

An ethics review will be needed to determine if the real-world driving data abides by all privacy requirements, e.g., are license plates and faces blurred?

Are the other agents in the sim scenarios purely replayed? Can they be controlled by CARLA's default agent behavior models to have a bit of agent interactivity? Or would that not work because of the introduction of errant behaviors (or something similar)?

**Details Of Ethics Concerns:**

An ethics review will be needed to determine if the real-world driving data abides by all data collection and privacy requirements, e.g., are license plates and faces blurred?

Also, concerns of data release and quality as no real-world data samples are provided for review.

---

### Official Review · Reviewer_CxpH · 2024-11-04

**Soundness:** 2
**Presentation:** 2
**Contribution:** 2
**Rating:** 3
**Confidence:** 5

**Summary:**

In this paper, the authors have introduced DriveE2E, a digital twin based benchmark embedded in Carla simulator for end-to-end autonomous driving. The benchmark includes 15 digital twin interaction creation, comprising of 800 distinct traffic scenarios. On the experimental side, the paper evaluate the performance of state-of-the-art end-to-end autonomous driving methods in their DriveE2E benchmark using success rate and driving score as evaluation metrics.

**Strengths:**

Here are the strengths of this paper:
1. Combining digital twin with Carla is promising to reduce the sim-to-real gap for end-to-end autonomous driving.

**Weaknesses:**

Following are the weakness of this paper:
1. The authors have claimed to address the sim-to-real gap for end-to-end autonomous driving, yet from experimentation it is hard to see any evidence for that. It would be good to see if the authors indicate the statistical analysis of how much they reduce the sim-to-real gap.
2. In the introduction, the authors indicate the motivation for developing this benchmark, but other benchmarks are available in the literature. The more close to their approach is the Interaction dataset and simulator. It would be good to see if the authors compare their benchmark with the existing benchmarks.
3. In the paper, figures such as Fig.3 and Fig.4 were not even mentioned in the text.
4. In the experiments, the authors have shown the performance of different E2E methods on their benchmark, but it would be good to see what is actually improved or what the implications derived from using their benchmark.
5. One more concern is that they have mentioned the city's name, which I think violates the double-blind review process.
6. I believe that in evaluating their benchmark, they need to do more experimentation since diffusion-based methods are now used in the literature for end-to-end autonomous driving that also generates the waypoints.
7. In the paper, the authors indicate that Carla uses rules to generate the dataset, but there is another method, Roach, that is Carla-based and uses RL to collect the expert data. It would be good if the authors also compared their approach with this one.

**Questions:**

Following are the weakness of this paper:
1. The authors have claimed to address the sim-to-real gap for end-to-end autonomous driving, yet from experimentation it is hard to see any evidence for that. It would be good to see if the authors indicate the statistical analysis of how much they reduce the sim-to-real gap.
2. In the introduction, the authors indicate the motivation for developing this benchmark, but other benchmarks are available in the literature. The more close to their approach is the Interaction dataset and simulator. It would be good to see if the authors compare their benchmark with the existing benchmarks.
3. In the paper, figures such as Fig.3 and Fig.4 were not even mentioned in the text.
4. In the experiments, the authors have shown the performance of different E2E methods on their benchmark, but it would be good to see what is actually improved or what the implications derived from using their benchmark.
5. One more concern is that they have mentioned the city's name, which I think violates the double-blind review process.
6. I believe that in evaluating their benchmark, they need to do more experimentation since diffusion-based methods are now used in the literature for end-to-end autonomous driving that also generates the waypoints.
7. In the paper, the authors indicate that Carla uses rules to generate the dataset, but there is another method, Roach, that is Carla-based and uses RL to collect the expert data. It would be good if the authors also compared their approach with this one.

---

### Note · Authors · 2024-11-14

I have read and agree with the venue's withdrawal policy on behalf of myself and my co-authors.